# Cytoskeleton Protein BmACT1 Is Potential for the Autophagic Function and Nuclear Localization of BmAtg4b in *Bombyx mori*

**DOI:** 10.3390/cells12060899

**Published:** 2023-03-15

**Authors:** Qiuqin Ma, Jianhao Deng, Hanbo Li, Zhijun Huang, Ling Tian

**Affiliations:** Guangdong Laboratory for Lingnan Modern Agriculture, Guangdong Provincial Key Lab of Agro-Animal Genomics and Molecular Breeding, College of Animal Science, South China Agricultural University, Guangzhou 510642, China

**Keywords:** BmAtg4a, BmAtg4b, autophagy, BmACT1, BmACT2

## Abstract

Homologs of Autophagy-related (Atg) protein 4 are reported to cleave LC3 protein and facilitate autophagy occurrence differently in mammals, whereas their functions have not been investigated in insects. Three homologs, including *BmAtg4a* and its short form *BmAtg4c* as well as *BmAtg4b,* exist in *Bombyx mori*. Herein, the autophagic functions of BmAtg4a and BmAtg4b were investigated. qPCR detection found that *BmAtg4a* and *BmAtg4b* both peaked during larval-pupal metamorphosis when autophagy occurs robustly. Immunofluorescent staining showed that BmAtg4a was predominantly localized at the cytoplasm, while BmAtg4b had notable nuclear localization. Overexpression of *BmAtg4a* and *BmAtg4b* both slightly promoted basal autophagy but inhibited the autophagy induced by the infection of *B. mori* nucleopolyhedrovirus (BmNPV) and, thereby, its proliferation. In comparison, knockout of *BmAtg4a* or *BmAtg4b* significantly upregulated BmNPV-induced autophagy and its replication in BmN cells. Results of Co-immunoprecipitation associated with mass spectrum showed that the cytoskeleton protein *B. mori* actin A2 (BmACT2) and *B. mori* actin A1 (BmACT1) bound with BmAtg4a and BmAtg4b especially. Knockout of *BmACT1* and *BmACT2* inhibited BmAtg4b- and BmAtg4a-induced autophagy, respectively; moreover, knockout of *BmACT1* reduced the ratio of cells with nuclear BmAtg4b. Of note, BmAtg4a and BmAtg4b had physical interaction, and they had an inhibitory effect on mutual autophagic function. In this work, we provide new insights into the autophagy machinery in insects as well as its function in the proliferation of BmNPV.

## 1. Introduction

Macroautophagy, hereafter referred to as autophagy, is a cellular degradation process to transport cytoplasmic material via double-membrane organelles called autophagosomes, which finally fuse with lysosomes and thereby leading to bulk degradation of the contents to recycle molecular materials for further development [1]. The maturation of autophagosomes is controlled by a series of protein complexes comprised of different Atg (autophagy-related) proteins. Autophagosome initiation is mediated by the ULK1/ATG1-ATG13 protein kinase complex, while the expansion and maturation are under the control of two ubiquitin-like conjugation systems (Atg8/LC3–PE and Atg5–Atg12-Atg16) [2,3]. Approximately 20 *Atg* genes have been identified in *Bombyx mori*, and most of them are transcriptional or post-translational regulated by the insect molting hormone 20-hydroxyecdysone in cooperation with nutrient signaling [4,5,6,7]. Moreover, autophagy plays critical roles in the response to pathogen invasion in insects [8]. The expressions of *Atg* genes are upregulated after the infection of *B. mori* nucleopolyhedrovirus (BmNPV) in BmN cells. Moreover, overexpression of *BmAtg7* and *BmAtg9* promotes autophagy and, thereby, the replication of BmNPV, showing a positive role of autophagy in viral proliferation [9].

ATG4 facilitates autophagy by promoting autophagosome maturation through reversible lipidation and delipidation of ATG8 [10]. In mammals, there are four ATG4 isoforms (ATG4A, ATG4B, ATG4C, and ATG4D), which are cysteine proteases containing catalytic and short-finger domains [11]. Moreover, the four ATG4 isoforms have different preferences for the members of the ATG8/LC3 family, ATG4B cleaves most LC3 isoforms, and LC3B is the favorite substrate, while ATG4C and ATG4D are both able to target GABARAP [1,12]. Moreover, ATG4C and ATG4D have the canonical DEVD (aspartic acid, glutamic acid, valine, and aspartic acid) sequence that is recognized by caspase and consequently involved in the apoptotic pathway [13]. Notably, ATG4 homologs undergo diverse post-translational modifications, which modulate the occurrence of autophagy in eucaryotes, including phosphorylation, acetylation, oxidation, S-nitrosylation, and ubiquitination [14,15].

The cytoskeleton proteins from the actin family are the most abundant protein in eukaryotic cells. They are highly conserved among different species and involved in multiple cellular processes such as cell movement, maintenance of cell shape, and transcriptional regulation [16,17]. In contrast, actin isoforms have different structures and thus function differently [18]. In breast and ovarian cancer cells, the actin skeleton facilitates the motility of cancer cells, which provides a basis for developing therapeutic agents to block cancer metastasis by targeting actin [19]. Besides, the movement of bacterial pathogens such as *Listeria* and *Shigella* is realized by the polymerization and movement of actin [20]. Invasion of the alphaherpesviruses leads to the remodeling of actin, which consequently promotes the migration of the alpha herpesvirus in the host cells [21]. Moreover, actin filaments are colocalized with ATG14, BECN1/Beclin1 and PtdIns3P-rich structures, and the depolymerization of actin abolishes the increase of autophagic vacuoles after starvation in mammalian cells [22]. In comparison, F-actin is disassembled in *ATG7*-knockout mouse embryonic fibroblasts (MEFs) during starvation [23]. However, the functions of actin are poorly understood in insects.

In *B. mori*, three homologs of Atg4 protein (BmAtg4a, BmAtg4b, and BmAtg4c, the short form of BmAtg4a) were reported in Genbank, while their precise functions have not been deeply investigated. Therefore, their functions in autophagy occurrence were studied. Notably, the cytoskeleton proteins are involved in BmAtg4-mediated autophagy, showing a new regulatory mechanism of autophagy and a new function of actin in insects.

## 2. Materials and Methods

### 2.1. Animals and Cell Lines

Silkworms (Dazao) were reared with fresh mulberry leaves at 25 °C under a 14 h light/10 h dark cycle [4]. The *B. mori* ovary cell line (BmN) was cultured in Grace’s insect Medium supplemented with 10% fetal bovine serum (AusGeneX, Gold Coast, Australia, FBS500-S) at 28 °C in a constant-temperature incubator. 

### 2.2. The Recombinant BmNPV-EGFP 

A full-length sequence of *Egfp* was cloned into a pFastBac Dual Expression Vector, and then the plasmid was transformed into DH10Bac competent cells integrated with the BmNPV genome [24]. Subsequently, the bacmid DNA was extracted and transfected into BmN cells to generate the P1 (progeny 1) virus, and then the P2 virus was obtained by infecting BmN cells with the P1 virus for 5 days; the TCID_50_ of a titer of P2 BmNPV was determined [25].

### 2.3. Immunofluorescent (IF) Staining 

The sterilized coverslips were placed in the 6-well plate during BmN cell plating. The full-length sequences of *BmAtg4a-His*, *BmAtg4b-HA*, *BmACT1-V5*, and *BmACT2-FLAG* were cloned and inserted into the overexpression vector pIEX4. The wild-type or gene-knockout cells were transiently transfected with *BmAtg4a-His*, *BmAtg4b-HA* or co-transfected with *BmAtg4a-His* and *BmACT2-FLAG* as well as *BmAtg4b-HA* and *BmACT1-V5* for 48 h. Then the cells were fixed in 4% paraformaldehyde for 20 min and thoroughly washed with PBS 3 times. Subsequently, the cells were blocked with QuickBlock™ blocking buffer for immunofluorescent staining (Beyotime, Shanghai, China, P0260) for 1 h at 4 °C and then incubated with primary antibodies against the tags including His (Cell Signaling Technology, Boston, MA, USA, 12698s; 1:200), FLAG (Cell Signaling Technology, Boston, MA, USA, 14793S; 1:200), HA (Cell Signaling Technology, Boston, MA, USA, 3724S; 1:200), V5 (Cell Signaling Technology, Boston, MA, USA, 13202s; 1:200) or BmAtg8 (ABclonal Biotechnology, Shanghai, China; 1:100) at 4 °C overnight. After being washed with PBS 3 times, the cells were further incubated with an Alexa Fluor-488-conjugated secondary antibody (Invitrogen, Waltham, MA, USA, A11008; 1:200) and/or Alexa Fluor-594-conjugated secondary antibody (Invitrogen, Waltham, MA, USA, A11012; 1:200) for 2 h at room temperature, which was co-stained with DAPI (Beyotime, Shanghai, China, C1005) for 30 min, and thoroughly washed with PBS for 3 times before observation. Images were obtained under a confocal microscope equipped with an Olympus digital camera (Olympus, Tokyo, Japan, FV3000). To quantify fluorescence-labeled BmAtg4b-HA and BmAtg8 puncta, images from three independent biological repeats were recorded. The percentage of cells stained with the nuclear signal of BmAtg4b was used to quantify the variation of subcellular localization of BmAtg4b.

### 2.4. LysoTracker Red Staining

The gene knockout or overexpressed BmN cells were stained with LysoTracker Red DND-99 dye at a final concentration of 50 nM (Thermo Fisher Scientific, Waltham, MA, USA, L7528) for 15 min at 37 °C and then washed with PBS 3 times. The samples were observed under an Olympus FV3000 confocal microscope. The acidified lysosomes were quantified by the density of LysoTracker staining in cells [5].

### 2.5. CRISPR/Cas9-Mediated Gene Knockout

The vectors pB-CRISPR and A3-Helper (piggyBac transposase expression vector) were used for gene knockout. The sgRNA for *BmAtg4a*, *BmACT2*, *BmAtg4b*, and *BmACT1* was designed and inserted into pB-CRISPR to construct the knockout vector (Appendix A). Subsequently, the plasmids of pB-CRISPR and A3-Helper were co-transfected into BmN cells [26]. BmN cells co-transfected with empty pB-CRISPR and A3-Helper vectors were used as control. After transfection for 48 h, 600 μg/mL zeocin (Gibco, Waltham, MA, USA, R25001) was added to the cells, which were chemically selected and maintained for 1 month to remove the cells unsuccessfully integrated with the CRISPR/Cas9 system in the genome to obtain a stable cell line [25].

### 2.6. Quantitative Real-Time PCR (qPCR)

Total RNA was extracted using TRIzol Reagent RNA (Vazyme, Nanjing, China, R401-01). cDNA was generated from mRNA by reverse transcription using a PrimeScript^TM^ RT reagent kit (Takara, Kyoto, Japan, RR047A). *Bmrp49* was used as the reference gene for qPCR detection using cDNA as a template, and *BmGAPDH* was used as the reference gene using genomic DNA as a template for detecting the replication of the viral genes [27]. The relative expression or gene copies was calculated by the 2^−ΔΔCt^ method as previously described [28]. The primer sequence of qPCR is shown in Appendix A.

### 2.7. Co-Immunoprecipitation (Co-IP) 

*BmAtg4a-His* and *BmAtg4b-HA*, *BmAtg4a-His* and *BmACT2-FLAG*, *BmAtg4b-HA* and *BmACT2-FLAG*, *BmAtg4a-His* and *BmACT1-V5*, as well as BmAtg4b-HA and BmACT1-V5, were co-overexpressed in BmN cells for 48h, and then the cells were harvested and lysed in NP-40 (Beyotime, shanghai, China; P0013F) lysis buffer supplemented with a complete protease inhibitor cocktail (Roche, Basel, Switzerland, 0469313201). The supernatant was incubated with His (Cell Signaling Technology, Boston, MA, USA, 12698S; 1:200), HA (Cell Signaling Technology, Boston, MA, USA, 3724S; 1:200), FLAG (Cell Signaling Technology, Boston, MA, USA, 14793S; 1:200) or V5 (Cell Signaling Technology, Boston, MA, USA, 13202s; 1:200) primary antibodies at 4 °C for 3–4 h, and then incubated with protein A/G agarose beads (Thermo Fisher, Waltham, MA, USA, 20421) overnight according to the standard procedure of Co-IP, immunoprecipitation by IgG was used as the negative control.

### 2.8. Western Blotting (WB) 

Western blotting was performed as previously described [25]. Primary antibodies were used to detect the protein levels of EGFP (TransGen, Beijing, China, HT801; 1:4000), His (Cell Signaling Technology, USA, 12698s; 1:3000), His (TransGen, Beijing, HT501; 1:4000), HA (Santa Cruz Biotechnology, Santa Cruz, CA, USA, sc-7392; 1:3000), FLAG (Sigma, USA, F1804; 1:3000), V5 (Sigma, Saint Louis, MO, USA, H1029; 1:3000), and BmAtg8 (ABclonal Biotechnology, Shanghai, China; 1:3000). Tubulin alpha (Beyotime, Shanghai, China, AT819; 1:5000) was used as the reference protein. All images of western blotting were taken with a Tanon-5200 Chemiluminescent Images System. Western blots of BmAtg8–PE and EGFP were quantified using ImageJ 1.6 software.

### 2.9. Immunoprecipitation Associated with Mass Spectrum

The immunoprecipitation of BmAtg4a-His and BmAtg4b-HA precipitated by the protein A/G agarose beads (Thermo Fisher, USA, 20421) were sent for mass spectrum analysis (Shanghai Luming Biotechnology Co., Ltd., Shanghai, China). Easy-nLC 1200 ultra-high performance liquid chromatograph equipped with a tandem high-resolution mass spectrometer (LC-MS) system was used for the analysis of proteins. Enzymatic digested proteins immunoprecipitated by BmAtg4a and BmAtg4b were loaded onto a reversed-phase analytical column (75 μm × 150 mm, RP-C18, ThermoFisher, Waltham, MA, USA, packed with PolySULPHOETHYL A, 5 μm, 200 Å, PolyLC Inc., Waltham, MA, USA) and separated by a gradient elution buffer. The peptides were identified using a tandem Q Exactive high-resolution mass spectrometer (ThermoFisher, Waltham, MA, USA). Proteins were further analyzed using ProteomeDiscover 2.5 software by searching the UniProt (*B. mori*) database. All the obtained proteins were subjected to COG (Clusters of Orthologous Groups) and KEGG (Kyoto Encyclopedia of Genes and Genomes) analysis.

### 2.10. Statistical Analysis

Each experiment was performed for three biological replicates. The statistical analysis of BmNPV *ie1*, *gp64*, and *helicase* was performed by unpaired Student’s *t*-test. The fluorescence comparisons between and control groups used unpaired Student’s *t*-test. * *p* < 0.05, ** *p* < 0.01, *** *p* < 0.001, n.s. no significance. To quantify the fluorescent BmAtg4b, viral EGFP, BmAtg8 puncta, and LysoTracker Red staining, about 300 BmN cells from 3 independent biological replicates were performed to record and analyze.

## 3. Results

### 3.1. Expression and Subcellular Localization of BmAtg4a and BmAtg4b

In mammals, four isoforms of ATG4 (ATG4A, ATG4B, ATG4C, and ATG4D) have been reported. In comparison, three homologs BmAtg4 (BmAtg4a, BmAtg4b, and BmAtg4c) proteins have been found in *B. mori*. Aligned by MegAlign 7.1 software, BmAtg4c (XP_021202571.1) was found to be a short form of BmAtg4a (XP_021202570.1), while the amino acid sequences of BmAtg4b (XP_037866680.1) were quite different with BmAtg4a and BmAtg4c (Appendix A). In order to reveal the precise functions of BmAtg4a and BmAtg4b, their expressions were detected in the fat body in the last larval instar from day 2 of the fifth instar (5L2D) to day 2 of prepupa (PP2) as well as their tissue tropism in the fat body, midgut, Malpighian tubule, silk gland, and sexual gland at 5L4D and PP2 stages. Results showed that mRNA levels of *BmAtg4a* and *BmAtg4b* were gradually increased from 5L2D to PP2 in *B. mori* fat bodies (Figure 1A). In addition, the expressions of *BmAtg4a* and *BmAtg4b* were both highest in sexual glands, followed by that in the midgut and Malpighian tubule at the 5L4D stage, while significantly increased in fat body but decreased in sexual glands at the PP2 stage (Figure 1B). Subsequently, the subcellular localization of BmAtg4a and BmAtg4b was detected by IF staining, and BmAtg4a was predominantly localized in the cytoplasm, while BmAtg4b existed in both the nucleus and cytoplasm in BmN cells (Figure 1C). Taken together, the expression of *BmAtg4a* and *BmAtg4b* are both consistent with autophagy occurrence in *B. mori*.

### 3.2. BmAtg4a and BmAtg4b Are Both Potential for Autophagy Occurrence 

Previous studies report that autophagy monitored by the protein level of BmAtg8–PE and LysoTracker staining is notably increased in *B. mori* fat body from 5L7D to PP2 when the larvae cease feeding and prepare for larval-pupal metamorphosis [29]. Therefore, the autophagic functions of *BmAtg4a* and *BmAtg4b* were investigated. As expected, overexpression of *BmAtg4a* and *BmAtg4b* both led to a significant increase of BmAtg8 puncta and caused notable lysosomal acidification indicated by LysoTracker Red staining in BmN cells (Figure 2A,A’,B,B’). Subsequently, *BmAtg4a* and *BmAtg4b* were co-overexpressed with *GFP-BmAtg8*, respectively, and WB results showed that their overexpression both slightly increased the protein levels of GFP-BmAtg8–PE formation (Figure 2C). In general, the overexpression of *BmAtg4a* and *BmAtg4b* promotes basal autophagy in *B. mori* BmN cells. 

In comparison, *BmAtg4a* (sg-*BmAtg4a*) or *BmAtg4b* (sg-*BmAtg4b*) was knockout mediated by CRISPR/Cas9 system in BmN cells. The knockout efficiencies of *BmAtg4a* and *BmAtg4b* were both near 100% (Appendix A). To our surprise, knockout of *BmAtg4a* or *BmAtg4b* dramatically upregulated the basal autophagy indicated by BmAtg8 punctation (Figure 2D,D’), LysoTracker Red staining (Figure 2E,E’), and WB of BmAtg8–PE formation (Figure 2F).

### 3.3. BmAtg4a and BmAtg4b Play Negative Roles in the Proliferation of BmNPV

A series of *Bombyx Atg* genes, which play positive roles in BmNPV replication, are transcriptionally affected by the infection of BmNPV in BmN cells [30]. However, the roles of *BmAtg4a* and *BmAtg4b* in viral proliferation suffer from a lack of investigation. After overexpression of *BmAtg4a* for 48 h, the BmN cells were infected with BmNPV (MOI = 5) for a further 48 h [25]. Monitored by gene copies of BmNPV *ie1*, *gp64*, and *helicase,* as well as the variation of the green fluorescence of recombinant BmNPV-EGFP, overexpression of *BmAtg4a* significantly reduced the viral replication (Figure 3A,B,B’). Consistently, WB of EGFP protein confirmed the inhibitory effects of *BmAtg4a* on BmNPV proliferation (Figure 3C). Of note, overexpression of *BmAtg4a* reduced the virus-induced BmAtg8–PE conjugation (Figure 3C), while the subcellular localization of BmAtg4a was not affected in BmN cells (Figure 3D). In comparison, knockout of *BmAtg4a* increased the gene copies of the viral *ie1*, *gp64*, and *helicase* (Figure 3E) as well as fluorescence and protein levels of BmNPV-expressed EGFP (Figure 3F,F’,G). Notably, knockout of *BmAtg4a* dramatically increased the virus-induced BmAtg8–PE conjugation (Figure 3G). 

Subsequently, the function of *BmAtg4b* in BmNPV proliferation was detected. Results showed that overexpression of *BmAtg4b* significantly reduced the replication and proliferation of BmNPV indicated by gene copies of the viral *ie1*, *gp64*, and *helicase* (Figure 4A), fluorescence and protein levels of virus-expressed EGFP (Figure 4B,B’,C). Accordingly, overexpression of *BmAtg4b* also partially abolished the virus-induced BmAtg8–PE conjugation but without affecting the subcellular localization of BmAtg4b (Figure 4C,D,D’). In comparison, knockout of *BmAtg4b* promoted the replication and proliferation of BmNPV (Figure 4E,F,F’,G). Moreover, knockout of *BmAtg4b* promoted BmNPV-induced conjugation of BmAtg8–PE (Figure 4G).

### 3.4. Identification of Proteins Involved in the Characteristics of BmAtg4a and BmAtg4b

*BmAtg4a* and *BmAtg4b* have similar autophagic functions but varied subcellular localization; thus, we next investigated the proteins involved in the formation of their characteristics. The overexpressed *BmAtg4a* and *BmAtg4b* were respectively immunoprecipitated by the antibody of fused His or HA tag (Appendix A), and then the immunoprecipitations were subjected to protein identification by mass spectrum. Results showed that there were 753 proteins existed in the immunoprecipitate of BmAtg4a and BmAtg4b in common, while 165 proteins bound with BmAtg4a and 331 proteins bound with BmAtg4b especially (Appendix A). The proteins immunoprecipitated by BmAtg4a and BmAtg4b were further analyzed by the Kyoto Encyclopedia of Genes (KEGG) and Cluster of Orthologous Groups of proteins (COG). KEGG functional annotation showed that BmAtg4a- and BmAtg4b-binding proteins were both involved in multiple biological processes, including translation, global and overview maps, transcription, folding, sorting and degradation, transport and catabolism, and signal transduction (Figure 5A, Appendix A). Moreover, COG annotation showed that the binding proteins were mainly distributed in functional systems such as translation, ribosomal structure and biogenesis, transcription, replication, recombination and repair, post-translational modification, protein turnover, chaperones, general function prediction only, and cytoskeleton (Figure 5B, Appendix A). Subsequently, two cytoskeleton proteins, BmACT2 and BmACT1, were found to bind with BmAtg4a and BmAtg4b specifically from the data of mass spectrum.

### 3.5. BmACT1 and BmACT2 Are Required for the Autophagic Functions of BmAtg4b and BmAtg4a

Results from the immunoprecipitation associated with mass spectrum suggested that the cytoskeleton proteins BmACT2 and BmACT1 were the specific binding proteins of BmAtg4a and BmAtg4b. Therefore, full-length sequences of *BmACT2* (NM_001126253.1) and *BmACT1* (NM_001126252.1) were cloned and fused with *FLAG* and *V5* tags, respectively. The interaction between the actin proteins BmACT2, BmACT1 and BmAtg4a/BmAtg4b was detected by co-immunoprecipitation (Co-IP), and results showed that BmAtg4a and BmACT2 in addition to BmAtg4b and BmACT1 had physical interactions, while BmAtg4a, BmACT1, and BmAtg4b and BmACT2 could not interact with each other, confirming the specific binding between BmACT2 and BmAtg4a as well as BmACT1 and BmAtg4b (Figure 6A). Of note, IF staining showed that neither *BmACT1* nor *BmACT2* overexpression affected the subcellular localization of BmAtg4b or BmAtg4a (Figure 6B,B’). In summary, BmACT1 and BmACT2 show specific protein binding with BmAtg4b and BmAtg4a, respectively.

In comparison, loss-of-function analysis of *BmACT1* or *BmACT2* was performed, and the knockout efficiencies of *BmACT2* and *BmACT1* were both near 100% mediated by CRISPR/Cas9 system in BmN cells (Appendix A). Compared to the control (pB-CRISPR), knockout of *BmACT1* (sg-*BmACT2*) and *BmACT2* (sg-*BmACT1*) didn’t affect the cytoplasmic localization of BmAtg4a (Figure 7A). At the same time, the knockout of *BmACT1* significantly reduced the percentage of cells with nuclear BmAtg4b, showing the requirement of BmACT1 for the nuclear localization of BmAtg4b (Figure 7B,B’). Subsequently, the effects of BmACT2 and BmACT1 on the autophagic function of BmAtg4a and BmAtg4b were further detected, and results showed that protein levels of BmAtg8–PE were reduced after overexpression of *BmAtg4a* in the *BmACT2*-knockout cells; similarly, the knockout of *BmACT1* downregulated BmAtg8–PE formation after overexpression of *BmAtg4b* (Figure 7C,D). In general, cytoskeleton proteins BmACT1 and BmACT2 are required for the autophagic function of BmAtg4b and BmAtg4a, respectively. Moreover, BmACT1 is vital for the nuclear localization of BmAtg4b.

### 3.6. BmAtg4a and BmAtg4b Inhibit Mutual Autophagic Functions

BmAtg4a and BmAtg4b function similarly in autophagy occurrence; thus, their interaction was investigated. As predicted by the software online (https://www.ebi.ac.uk/msd-srv/prot_int/website, accessed on 26 October 2021), the putative structure of BmAtg4a had physical binding with that of BmAtg4b. The alpha helix of BmAtg4a contained 141 amino acids, and the beta-turn was comprised of 16 amino acids; in addition, the random coil contained 216 amino acids. The alpha helix of BmAtg4b harbored 118 amino acids, and the beta-turn had 16 amino acids. Moreover, the random coil had 188 amino acids. There existed five predicated hydrogen bonding sites (between BmAtg4a S 72 and BmAtg4b G 105, BmAtg4a S 75 and BmAtg4b S 34, BmAtg4a S 75 and BmAtg4b K 36, BmAtg4a E 78 and BmAtg4b T 33, and BmAtg4a S 68 and BmAtg4b Q 106) and one salt bridge site (between BmAtg4a E 53 and BmAtg4b K 102) between BmAtg4a and BmAtg4a (Figure 7E and Appendix A). Therefore, the interaction between BmAtg4a and BmAtg4b was further verified by Co-IP, and results showed that only BmAtg4b could pull down BmAtg4a (Figure 7F). Subsequently, the effects of BmAtg4a and BmAtg4b on mutual autophagic function were investigated. Of note, overexpression of *BmAtg4a* in the *BmAtg4b*-knockout cells significantly increased the protein levels of BmAtg8–PE (Figure 7G and Appendix A). Similarly, overexpression of *BmAtg4b* also dramatically promoted BmAtg8–PE conjugation in the *BmAtg4a*-knockout cells, showing the upregulation of autophagy in the absence of the other homolog (Figure 7H and Appendix A).

In general, there are mainly two BmAtg4 homologs in *B. mori*, which function similarly in autophagy occurrence. The cytoskeletal protein BmACT2 and BmACT1 are specifically binding with BmAtg4a and BmAtg4b. Moreover, they are involved in BmAtg4a- or BmAtg4b-mediated autophagy. In addition, BmACT1 has potential for the nuclear localization of BmAtg4b. Of note, knockout of *BmAtg4a* and *BmAtg4b* enhanced mutual autophagic function, suggesting the inhibitory role of the heterodimer of BmAtg4a and BmAtg4b in autophagy occurrence.

## 4. Discussion

ATG4 catalyzes proteolytic cleavage of ATG8 to facilitate its subsequent targeting of PE during autophagy occurrence. Reports show that different isoforms of ATG4 are usually associated with the occurrence of diverse diseases. In chronic myeloid leukemia cells, the knockdown of *ATG4B* suppresses autophagy and reduces the survival of cells, which increases the sensitivity of cancer cells to chemotherapy treatment [31,32]. *ATG4A* and *ATG4C* are required for the survival of breast cancer stem cells or mammospheres formation, showing their important roles in the progress of breast cancer [33,34]. Besides, women who carry a variant allele of *ATG4A* face a lower risk of ovarian cancer [35]. Meanwhile, ATG4D was considered a tumor suppressor in colorectal carcinogenesis [36]. Of note, lacking *ATG4B* causes a severe but incomplete defect in LC3/GABARAP lipidation and autophagy in human HAP1 and HeLa cells. Moreover, loss-of-function analysis of ATG4 isoforms reveals that each of ATG4A, ATG4B and ATG4D has residual priming activity, which is sufficient to enable lipidation of endogenous GABARAPL1 on autophagic structures, showing the redundant functions of the ATG4 isoforms in autophagy occurrence [1]. In contrast, the autophagic functions of Atg4 homologs are poorly understood in insects. 

There are two main Atg4 homologs in *B. mori* named BmAtg4a and BmAtg4b. Overexpression of *BmAtg4a* and *BmAtg4b* mildly induced basal autophagy in BmN cells. To our surprise, knockout of *BmAtg4a* and *BmAtg4b* both significantly promoted autophagy, especially the lipidation of BmAtg8, suggesting the limitation of basal autophagy in the co-existence of the two BmAtg4 proteins. Subsequently, we found that BmAtg4b was able to interact with BmAtg4b physically. Knockout of *BmAtg4a* promoted autophagy after *BmAtg4b* overexpression; similarly, knockout of *BmAtg4b* facilitated BmAtg4a-induced conjugation of BmAtg8–PE. These data indicated that the heterodimer between BmAtg4a and BmAtg4b partially abolished mutual function in processing the lipidation of BmAtg8, which was well consistent with the results that their single knockout promoted basal autophagy. In our previous study, the knockdown of BmAtg4 significantly blocks the 20E-regulated autophagy in the fat body during the larval-pupal transition [29]. This inconsistency indicates that the roles of BmAtg4a and BmAtg4b in different tissues or conditions will be varied, but they seem to be redundant in the occurrence of autophagy, which is similar to their mammalian homologs. 

Autophagy is activated after infection of the baculovirus BmNPV, which is, in turn, utilized by the virus for self-proliferation in *B. mori* [37]. It has been reported that the BmAtg8 protein interacts with the polyhedrin proteins of BmNPV directly, and they are colocalized on autophagosome membranes in BmN cells [38]. Whereas overexpression of the *BmAtg7*, *BmAtg9*, and *BmAtg13* promotes the expression of BmNPV genes and increases viral titer after the virus infection for 24 h in BmN-SWU1 cells [9,39]. Herein, we found that overexpression of *BmAtg4a* and *BmAtg4b* inhibited, while their knockout promoted the proliferation of BmNPV as well as the virus-induced autophagy after viral infection for 48 h. In general, autophagy plays positive roles in BmNPV replication in our’s and previous studies. In contrast, the two homologs of BmAtg4 display inhibitory effects on viral proliferation and virus-induced autophagy for a long period after infection, providing insights into the new strategy for anti-BmNPV research.

Notably, although the autophagic function was similar between BmAtg4a and BmAtg4b, their subcellular localization was quite different: BmAtg4a predominantly existed in the cytoplasm, while BmAtg4b had significant nuclear localization. The F-actin skeleton is involved in the transportation of membrane vesicles from different cellular compartments to autophagosomes and then enables the fusion of autophagosomes and lysosomes in mammals [17,40]. Immunoprecipitation associated with the mass spectrum and further identification found that BmACT1 and BmACT2 were the cytoskeleton proteins specifically binding with BmAtg4b and BmAtg4a and played key roles in BmAtg4b- or BmAtg4a-induced autophagy, showing the involvement of actin in autophagy occurrence in *B. mori*. Notably, BmACT1 was required for the nuclear localization of BmAtg4b. It is worth noting that deacetylation of a series of Atg proteins such as *Bombyx* and human Atg4, Atg8/LC3 lead to their nucleo-cytoplasm translocation and further promotion of autophagy [15,29,41]. Meanwhile, the precise mechanism of how autophagy is upregulated in the cytoplasm is not well documented. Herein, we hypothesize that the cytoplasmic BmACT1 might affect the autophagic function and nuclear localization of BmAtg4b in *B. mori* by mediating its transportation to the cellular membrane structure to form autophagosomes or to the location where BmAtg4b is acetylated and thereby nuclear translocation, but how these are realized is unclear. In addition, the molecular functions of the actin family remain largely unknown in insects, and these are worthy of investigation.

## 5. Conclusions

In conclusion, BmAtg4a and BmAtg4b are both positive for the occurrence of basal autophagy, while they have an inhibitory effect on mutual autophagic function and BmNPV proliferation through physical binding. Besides, the cytoskeleton proteins BmACT1 and BmACT2, which are potential for BmAtg4b- and BmAtg4a-induced autophagy, are found to be specifically binding with BmAtg4b and BmAtg4a. Of note, BmACT1 is also required for the nuclear localization of BmAtg4b. These findings shed lights on autophagy machinery in insects as well as its function in the proliferation of BmNPV.

## Figures and Tables

**Figure 1 cells-12-00899-f001:**
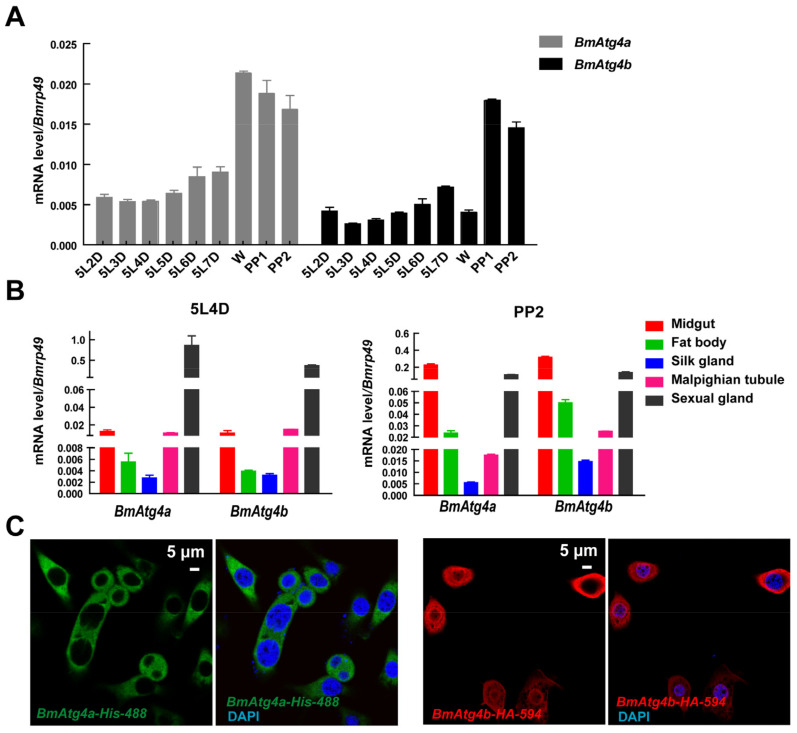
Detection of *BmAtg4a* and *BmAtg4b* expression as well as the subcellular localization of their proteins. (**A**) Developmental profiles of *BmAtg4a* and *BmAtg4b* from 5L2D to PP2 in *B. mori* fat body detected by qPCR. (**B**) mRNA levels of *BmAtg4a* and *BmAtg4b* in the fat body, midgut, Malpighian tubule, silk gland, and sexual gland at the 5L4D and PP2 stages. (**C**) The subcellular localization of the overexpressed BmAtg4a and BmAtg4b proteins in BmN cells was detected by IF staining.

**Figure 2 cells-12-00899-f002:**
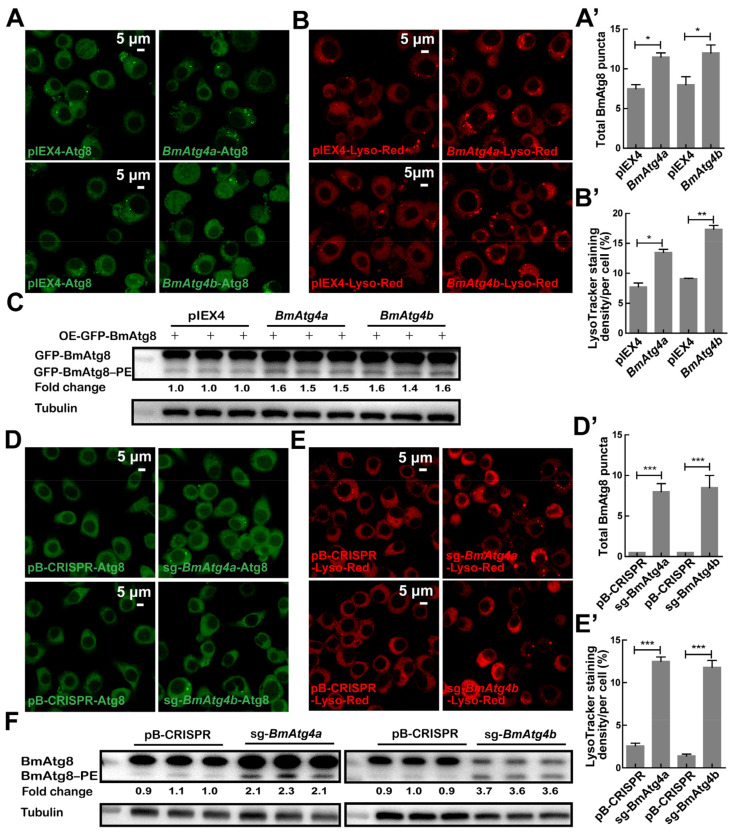
Autophagic function of BmAtg4a and BmAtg4b in BmN cells. (**A**) Punctation of BmAtg8 after overexpression of *BmAtg4a* and *BmAtg4b* detected by IF staining. (**A’**) Quantification of BmAtg8 puncta in (**A**). (**B**) Lysosomal acidification detected by LysoTracker Red staining after overexpression of *BmAtg4a* or *BmAtg4b*. (**B’**) Quantification of LysoTracker Red staining in (**B**). (**C**) After co-overexpression with *BmAtg4a* or *BmAtg4*, the protein levels of GFP-BmAtg8–PE were detected by WB using EGFP primary antibody. Overexpression of the pIEX4 empty vector was used as a control. Fold change means the variation of GFP-BmAtg8–PE protein levels in (**C**). (**D**) Punctations of BmAtg8 after knockout of *BmAtg4a* and *BmAtg4b* were detected by IF staining. (**D’**) Quantification of BmAtg8 puncta in (**D**). (**E**) Lysosomal acidification detected by LysoTracker Red staining after knockout of *BmAtg4a* or *BmAtg4b*. (**E’**) Quantification of LysoTracker Red staining in (**E**). (**F**) WB of endogenous BmAtg8–PE conjugation after knockout of *BmAtg4a* or *BmAtg4b*, and tubulin was used as the reference protein. Cells knockout by the empty pB-CRISPR vector was used as a control. Fold change means the variation of BmAtg8–PE protein levels in (**F**). * *p* < 0.05, ** *p* < 0.01, *** *p* < 0.001.

**Figure 3 cells-12-00899-f003:**
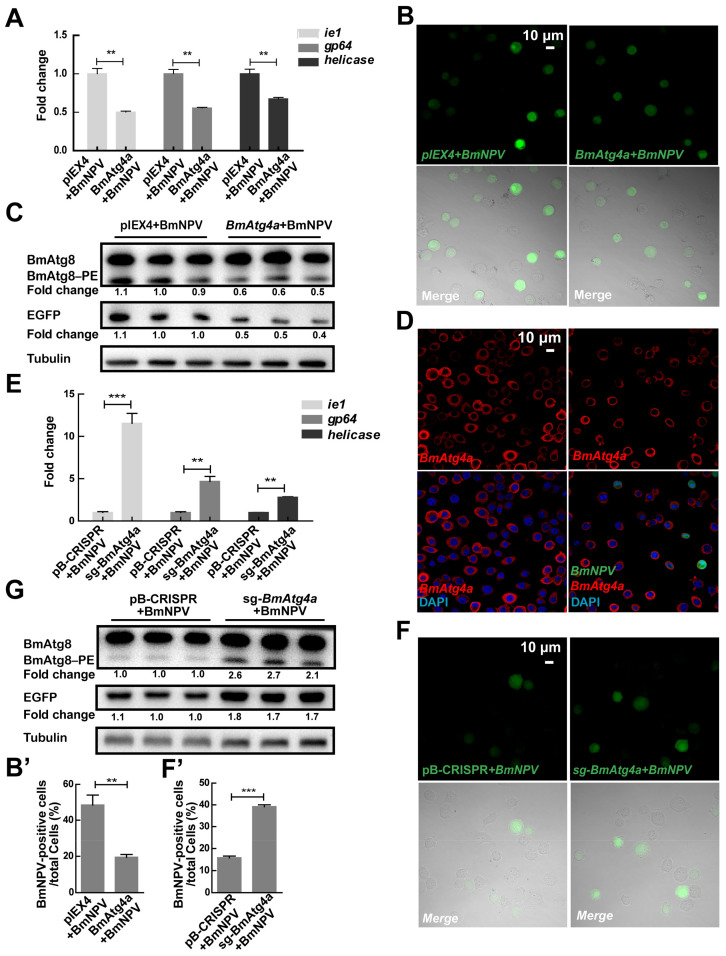
Effect of BmAtg4a on the proliferation of BmNPV in BmN cells. (**A**) The copies of BmNPV genes, including *ie1*, *gp64*, and *helicase,* were detected by qPCR after overexpression of *BmAtg4a* for 48 h, followed by BmNPV infection for a further 48 h. (**B**) The green fluorescence of BmNPV-EGFP was observed after the overexpression of *BmAtg4a*. (**B’**) Quantification of the EGFP-positive cells in (**B**). (**C**) The protein levels of BmAtg8/BmAtg8–PE and the viral EGFP detected by WB after overexpression of *BmAtg4a*. Fold change means the variation of BmAtg8–PE and EGFP protein levels in (**C**). (**D**) IF staining of the overexpressed BmAtg4a after BmNPV infection for 48 h. (**E**) The gene copies of *ie1*, *gp64*, and *helicase* were detected by qPCR after the knockout of *BmAtg4a*. (**F**) The green fluorescence of BmNPV-EGFP was observed after the knockout of *BmAtg4a*. (**F’**) Quantification of EGFP-positive cells in (**F**). (**G**) The protein levels of BmAtg8/BmAtg8–PE and the virus-expressed EGFP were detected by WB after knockout of *BmAtg4a*. Fold change indicates the variation of BmAtg8–PE and EGFP protein levels in (**G**). ** *p* < 0.01, *** *p* < 0.001.

**Figure 4 cells-12-00899-f004:**
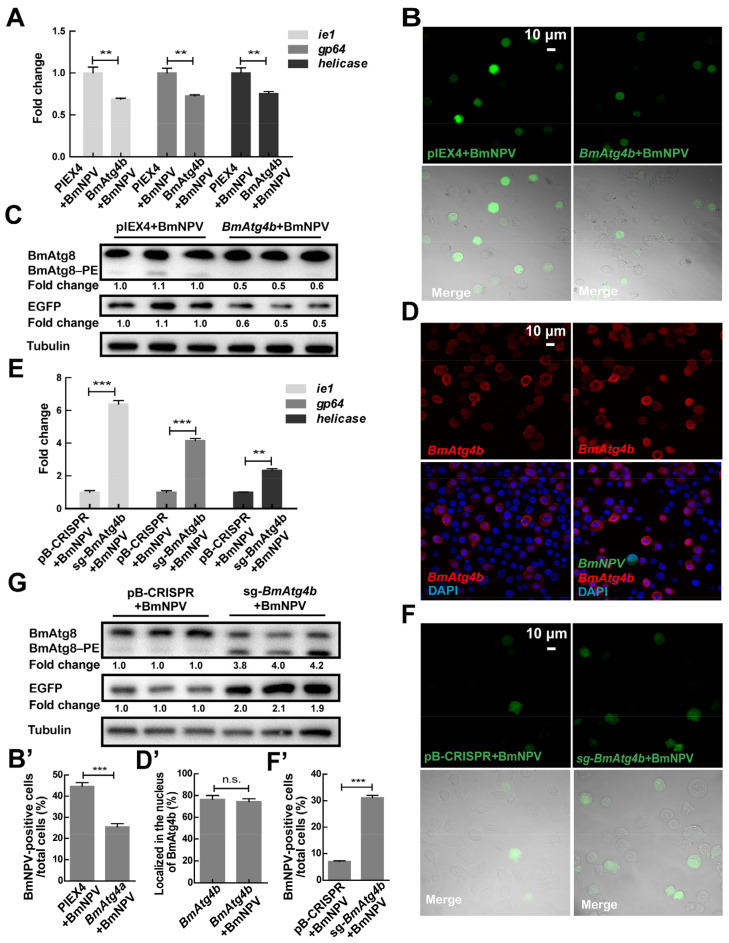
Effect of BmAtg4b on the proliferation of BmNPV in BmN cells. (**A**) The copies of BmNPV genes, including *ie1*, *gp64*, and *helicase,* were detected by qPCR after overexpression of *BmAtg4b* for 48 h, followed by BmNPV infection for a further 48 h. (**B**) The green fluorescence of BmNPV-EGFP was observed after the overexpression of *BmAtg4b*. (**B’**) Quantification of the EGFP-positive cells in (**B**). (**C**) The protein levels of BmAtg8/BmAtg8–PE and the viral EGFP detected by WB after overexpression of *BmAtg4b*. Fold change means the variation of BmAtg8–PE and EGFP protein levels in (**C**). (**D**) IF staining of the overexpressed BmAtg4b after BmNPV infection for 48 h. (**D’**) Quantification of cells with the nuclear signal of BmAtg4b in (**D**) indicated by the percentage of cells stained with the BmAtg4b signal in the nuclei. (**E**) The copies of *ie1*, *gp64*, and *helicase* were detected by qPCR after knockout of *BmAtg4b*, followed by BmNPV infection for a further 48 h. (**F**) The green fluorescence of BmNPV-EGFP was observed after the knockout of *BmAtg4b*. (**F’**) Quantification of the EGFP-positive cells in (**F**). (**G**) The protein levels of BmAtg8/BmAtg8–PE and the virus-expressed EGFP detected by WB after knockout of *BmAtg4b*. Fold change indicates the variation of BmAtg8–PE and EGFP protein levels in (**G**). ** *p* < 0.01, *** *p* < 0.001, n.s.: no significance.

**Figure 5 cells-12-00899-f005:**
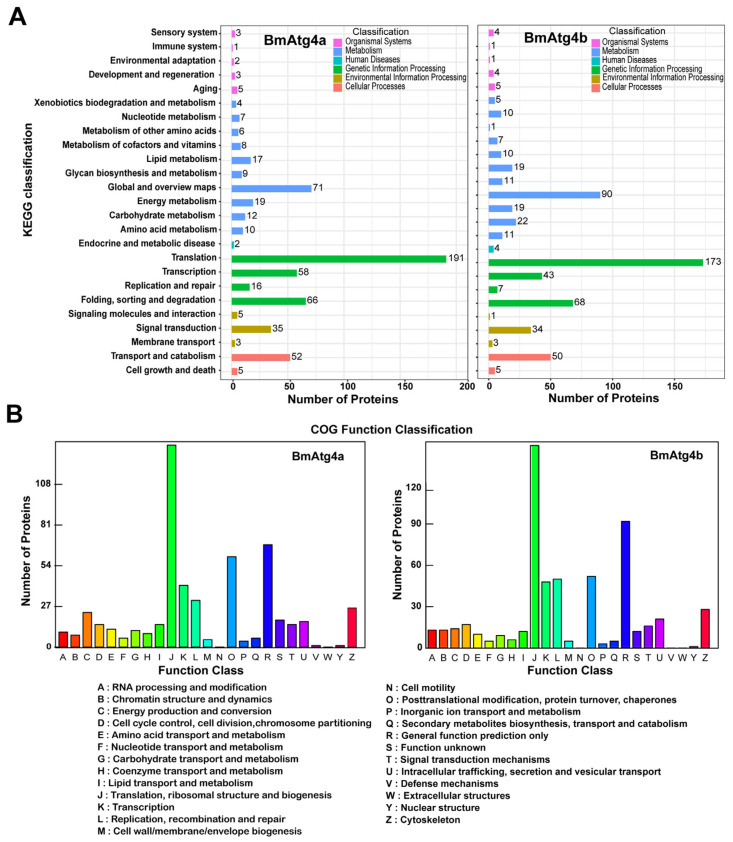
KEGG and COG analysis of BmAtg4a- and BmAtg4b-immunoprecipitated proteins. KEGG functional annotation (**A**) and COG annotation (**B**) of the proteins identified by mass spectrum in the immunoprecipitate of BmAtg4a-His and BmAtg4b-HA.

**Figure 6 cells-12-00899-f006:**
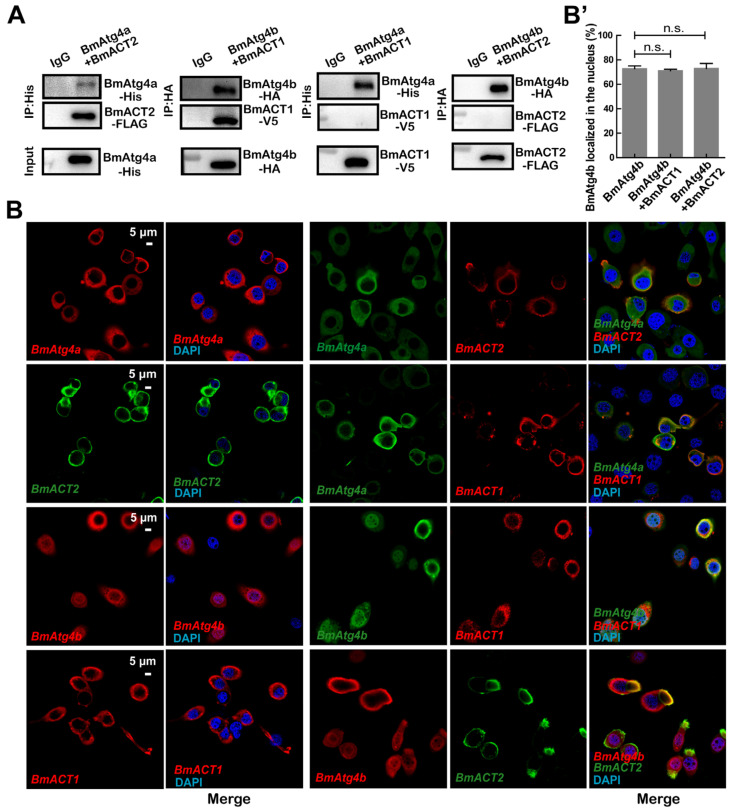
Interaction between BmAtg4a/BmAtg4b and BmACT2/BmACT1 indicated by Co-IP and IF staining in BmN cells. (**A**) Co-IP assays of BmAtg4a and BmACT2, BmAtg4b and BmACT1, BmAtg4a and BmACT1, as well as BmAtg4b and BmACT2. (**B**) IF staining of the co-overexpressed BmAtg4a and BmACT2, BmAtg4b and BmACT1, BmAtg4a and BmACT1 as well as BmAtg4b and BmACT2. (**B’**) Quantification of nuclear BmAtg4b in (**B**) indicated by the percentage of cells stained with BmAtg4b signal in the nuclei. n.s.: no significance.

**Figure 7 cells-12-00899-f007:**
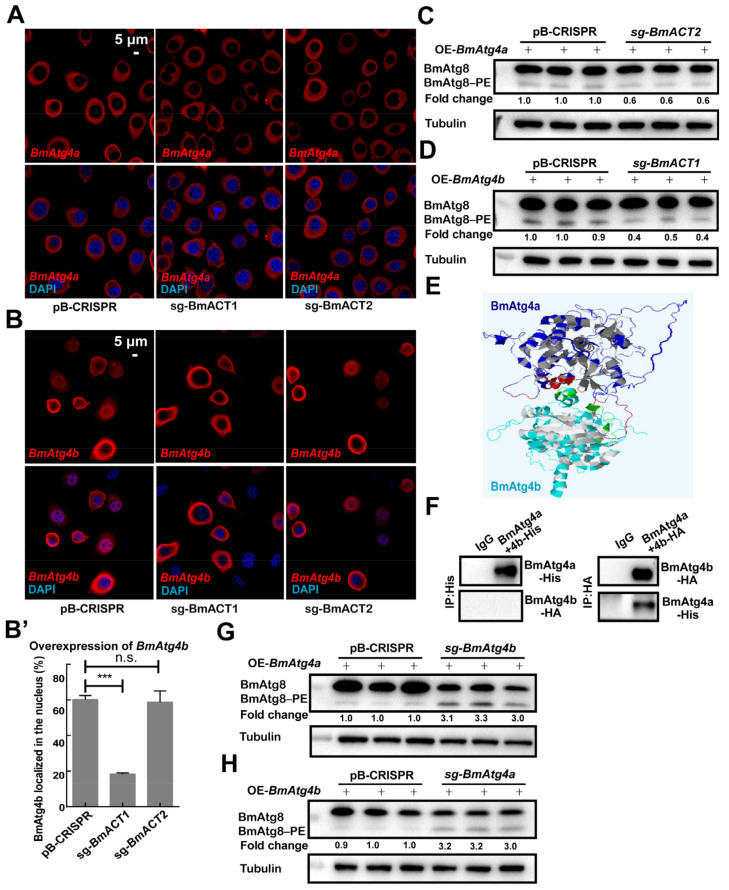
IF staining and autophagic functions of BmAtg4a and BmAtg4b after Knockout of *BmACT2* and *BmACT1* in BmN cells. (**A**) IF staining of the overexpressed *BmAtg4a* in the *BmACT2*- or *BmACT1*-knockout cells. (**B**) IF staining of the overexpressed *BmAtg4b* in the *BmACT2*- or *BmACT1*-knockout cells. (**B’**) Quantification of nuclear BmAtg4b in (**B**) indicated by the percentage of cells stained with BmAtg4b signal in the nuclei. (**C**) Protein levels of BmAtg8/BmAtg8–PE after overexpression of *BmAtg4a* in the *BmACT2*-knockout cells. (**D**) Protein levels of BmAtg8/BmAtg8–PE after overexpression of *BmAtg4b* in the *BmACT1*-knockout cells. (**E**) Predicated binding between BmAtg4a and BmAtg4b based on their putative structures. (**F**) Co-IP assay performed between BmAtg4a and BmAtg4b. (**G**) Protein levels of BmAtg8/BmAtg8–PE after overexpression of *BmAtg4a* in the *BmAtg4b*-knockout cells. (**H**) Protein levels of BmAtg8/BmAtg8–PE after overexpression of *BmAtg4b* in the *BmAtg4a*-knockout cells. Fold change indicates the variation of BmAtg8–PE protein levels in (**C**,**D**,**G**,**H**). *** *p* < 0.001, n.s.: no significance.

## Data Availability

All study data are included in this article. Where no new data were created.

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
