# Peer review of "Cytoskeleton Protein BmACT1 Is Potential for the Autophagic Function and Nuclear Localization of BmAtg4b in Bombyx mori"

_cells, 2023, doi:10.3390/cells12060899_

Round 1

Reviewer 1 Report

Review of Qiuqin Ma et al’s manuscript. The manuscript title is: "Cytoskeleton protein BmACT1 is potential for the autophagic function and nuclear localization of BmAtg4b in Bombyx mori".

The main aim of the manuscript is to compare the expression and function of two Atg4 homologs (BmAtg4a and BmAtg4b) in Bombyx mori model organisms. Their results suggest that both BmAtg4a and BmAtg4b are required for autophagy and regulation of viral replication. Their results suggest that the proteins can physically interact and inhibit each other. BmAtg4a and BmAtg4b have partially different subcellular localization, which may indicate moderately different functions. Their experiments have been complemented by valuable protein interaction studies.

The experiments presented are of high quality and the results are convincing.  However, in my opinion, further experiments would be necessary to interpret the results and I propose a major revision.  If my questions are answered adequately, I consider the manuscript suitable for publication in the journal Cells.

Main comments:

1. The nuclear localization of BmAtg4b is overemphasized in the text. Figure 1 C and later figures show that BmAtg4b is also found in the cytoplasm. it would be worthwhile to modify the modalities for localization in the text accordingly.

2. BmAtg4a KO showed a dramatic increase in virus-induced Atg8a-PE levels. What happened to the expression of Atg4b and -c homologs in the BmAtg4a mutants?

3. Atg4 regulates Atg8 protein at two sites during autophagosome maturation. It is required for maturation prior to lipidation of Atg8 (Atg4 cleaves Atg8 C-terminally) and is also required for cleavage from the outer membrane of the mature autophagosome. The amount of Atg8-PE in a sample can vary as Atg8 lipidation changes, but Atg4 is also affected by the efficiency of degradation. This is because Atg8-PE is a substrate for autophagy on the inner membrane of the autophagosome. Thus, an increased level of Atg8-PE in the absence of other autophagy substrates may indicate either activation of autophagy or inhibition following Atg8 lipidation. It seems necessary to decide, in the western blot results (especially Figure 2), how Atg8-PE levels change independently of degradation. It would be unwise to inhibit acid degradation with Bafilomycin-A because the level of Atg8-PE in Atg4 mutant and overexpressed samples would only vary as a function of the steps before lysosomal degradation.

4. How can the absence of BmAtg4a and BmAtg4b explain the increased Atg8-PE levels? How is Atg8 cleaved upstream of the PE binding? Is it possible that Atg4 homologues have redundant functions?

Minor comments:

1. In Figure 2 C, it would be useful to mark the GFP-BmAtg8a-PE band.

2. It is recommended to define "sg" for sg-BmAtg4a and -b in the figure signatures.

Reviewer 2 Report

The manuscript presents important data on two autophagy-related proteins of Bombyx mori, concerning their role in the process, their involvement in BmNPV infection, as well as describes the interaction profile of these proteins with two cytoskeleton proteins including the impact of such interaction on autophagy. Most of the results are robust, consistent, and well-presented. Some important questions and suggestions are presented below.    

Minor comments:

Many sentences from the abstract must have English Language revised.

Line 66: English language should be revised.

Line 73: the first part of the sentence is poorly written (makes no sense) and the second part appears to me as a very general statement wrongly linked to a specific effect of actin depolymerization.

Line 79: NCBI? Genbank? Please specify further.

Line 101 Apparently the term Bmg4-His is repeated.

Line 177: Replace “imagines” for “images”.

Line 185: The sentence needs to be rewritten.

Line 197: I don't think that statement allows including the expression in the sexual glands, because in this site, it seems to be the opposite.

Line 212: The sentence needs to be rewritten.

Figure 2C: bands correspond to GFP-BmAtg8-PE? PE is not indicated. Is it similar to figure 2F?

The legends of figures 3D and 4D must be better detailed.

Linha 303: binding-proteinS.

Line 367: add “respectively” after “BmAtg4b” and correct the word mediated.

Is it possible to make this causal relation among the results presented in the sentence from line 387 (comparing breast cancer with ovarian cancer)?

Line 402: correct the phrase: “Of note, BmAtg4b physically interacted with BmAtg4b,”.

Major comments:

Abstract must be more concise, presenting the results in a more global way, linked to each other. As it is, the results are presented only occasionally, which makes the text excessively complex and uninteresting for the reader.

Details on how fluorescence was quantified should be better described. Description presented on lines 115 – 117 is not sufficient. This aspect is especially important when it comes to the assertion of authors regarding the cellular localization of BmAtg4b: “BmAtg4b was mainly distributed in the nucleus in BmN cells” (line 200 and others). The image presented definitely does not represent this assertion. It is, in fact, clear that some positive reaction is observed in the nucleus, but it is not the main localization of this protein. This result has an important role in the results. Thus, may the authors comment on that?

In WB reactions, all bands that showed difference in expression should be quantified, as these are results are fundamental data for the overall discussion. 

Round 2

Reviewer 1 Report

Dear Editor,

I have received adequate and thorough answers to my questions from the authors. The revised manuscript has been significantly improved. Based on these, I find the manuscript to be suitable for publication.

Reviewer 2 Report

After the modifications, the manuscript can be accepted in its current form.